# OpenMathInstruct-1: A 1.8 Million Math Instruction Tuning Dataset

**Shubham Toshniwal, Ivan Moshkov, Sean Narenthiran, Daria Gitman,
Fei Jia, Igor Gitman**

NVIDIA

## Abstract

Recent work has shown the immense potential of synthetically generated datasets for training large language models (LLMs), especially for acquiring targeted skills. Current large-scale math instruction tuning datasets such as MetaMathQA [1] and MAmmoTH [2] are constructed using outputs from closed-source LLMs with commercially restrictive licenses. A key reason limiting the use of open-source LLMs in these data generation pipelines has been the wide gap between the mathematical skills of the best closed-source LLMs, such as GPT-4, and the best open-source LLMs. Building on our proposed prompting novelty, the recent progress in open-source LLMs, and some brute-force scaling, we construct OpenMathInstruct-1, a high-quality math instruction tuning dataset with 1.8M problem-solution pairs. The dataset is constructed by synthesizing *code-interpreter* solutions for GSM8K and MATH, two popular math reasoning benchmarks, using the recently released and permissively licensed Mixtral model. Our best model, OpenMath-CodeLlama-70B, trained on a subset of OpenMathInstruct-1, achieves a score of 84.6% on GSM8K and 50.7% on MATH, which is competitive with the best *gpt-distilled* models. To support the open-source efforts, we have released our code, models, and the OpenMathInstruct-1 dataset under a commercially permissive license.[1]

## 1 Introduction

The huge development and inference costs associated with general-purpose large language models (LLMs) have led to the rise of smaller, task-specific LLMs. Recent work has proposed creating these domain/task-specific LLMs by generating *high-quality synthetic data* using powerful closed-source models such as GPT-3.5/4 [3] and training smaller models on the generated *distillation* data [4, 5, 6]. For mathematical reasoning, our task of interest, all the current state-of-the-art open-source models are *gpt-distilled* [7, 2, 8, 9]. However, model development recipes relying on proprietary models like GPT-4 can have serious limitations: (a) legal restraints on how the finetuned models can be used,[2] (b) generating data with closed-source models is typically costlier than state-of-the-art open-source models, and (c) these recipes lack reproducibility as closed-source model behaviors can vary significantly over time [10].

*For developing mathematical reasoning models, why are open-source models not used in place of closed-source models?* To answer this, we compare GPT-4 with the Mixtral 8x7B model [11], one of the best open-source LLMs at mathematical reasoning in early 2024, by generating *code-interpreter* style solutions for two popular mathematical reasoning benchmarks, namely GSM8K [12]

---

[1]Data and models are available at `https://huggingface.co/collections/nvidia/openmath-65c5619de2ba059be0775014`
Code is available at `https://github.com/NVIDIA/NeMo-Skills`

[2]`https://openai.com/policies/terms-of-use`

38th Conference on Neural Information Processing Systems (NeurIPS 2024) Track on Datasets and Benchmarks.

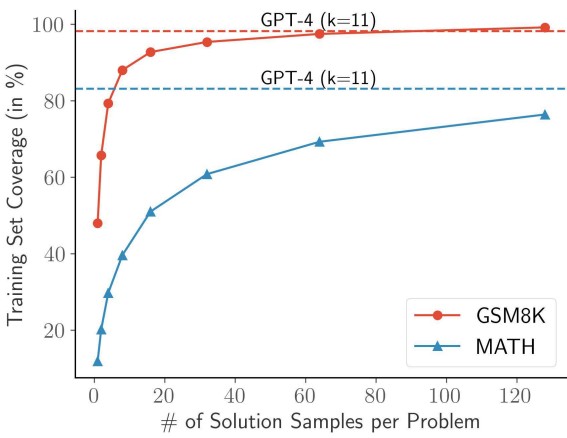

Figure 1: Training set coverage of Mixtral model generated solutions as a function of number of solutions sampled per problem (using temperature of 1.0 and top_p = 0.95). The statistics for the training set coverage of GPT-4 are from [8].

and MATH [13]. We use the metric *training set coverage (TSC)* to compare the models, where TSC measures the number of training problems for which any of the generated solutions leads to the ground truth answer (pass@k). Figure 1 shows the training set coverage (TSC) of the Mixtral model as a function of the number of sampled solutions. For the relatively easier GSM8K benchmark, the Mixtral model's coverage catches up to GPT-4's with almost 8x the number of solution samples. For the challenging MATH benchmark, even with 12x the number of solutions, the Mixtral model still has a lower TSC than GPT-4. This gap in the training set coverage reflects the distillation data quality and, hence, the quality of the final fine-tuned model. This explains the preference for GPT-4 in the current distillation pipelines for mathematical reasoning.

*Bridging the coverage gap between GPT-4 and Open-source LLMs*: We limit our investigation of open-source LLMs for synthesizing solutions to the Mixtral-base model due to (a) its strong performance on mathematical reasoning tasks compared to other open-source LLMs, and (b) its permissive license.[3] As a first attempt, we use a brute-force approach of sampling several solutions per problem. However, this approach only scales logarithmically, limiting its effectiveness (Figure 1). Next, we explore the approach of targeted solution generation, where we write few-shot prompts focused on specific sections of the training data. Concretely, we write few-shot prompts for each mathematics subject in the MATH dataset and merge the synthesized solutions. The motivation is that these subject-specific few-shot prompts could better target the latent mathematical capabilities of these general-purpose LLMs. Unfortunately, we only find a marginal gain in TSC with this approach (Section 2.2.2). Finally, we utilize the fact that reference text solutions accompany mathematical benchmarks such as MATH and GSM8K. These reference solutions can aid the synthesis of code-interpreter style solutions. We show that using these reference solutions in our few-shot prompt with a slight modification substantially increases the coverage and, consequently, the performance of the fine-tuned model (Section 2.2.3 and 4.1.2).

Our solution synthesis experiments result in OpenMathInstruct-1, a collection of 1.8M problem-solution pairs. OpenMathInstruct-1 has a training set coverage of 93% for MATH and 99.9% for GSM8K. Table 1 shows that compared to previous mathematical reasoning fine-tuning datasets, OpenMathInstruct-1 is at least four times bigger, and, even more importantly, it is permissively licensed, allowing unrestricted usage by future work. To illustrate the quality of OpenMathInstruct-1, we train and release a range of models based on Mistral-7B [14], Llama 2 [15], and CodeLlama [16]. In particular, the CodeLlama-70B model fine-tuned on a subset of OpenMathInstruct-1, referred to as OpenMath-CodeLlama-70B, achieves a score of 84.6% on GSM8K and 50.7% on MATH. These scores are competitive with the current best *GPT-distilled* models. Finally, to support the open-source efforts in this direction, we have publicly released all our fine-tuned models, code, and the OpenMathInstruct-1 dataset, along with a further 6.6M incorrect sampled solutions under a commercially permissive license.[4]

---

[3] https://mistral.ai/news/mixtral-of-experts/
[4] The incorrect solution trajectories can be used to train verifier models [12, 17, 18].

Table 1: Comparison of OpenMathInstruct-1 with mathematical reasoning fine-tuning datasets used by current state-of-the-art open-source models. OpenMathInstruct-1 is 4x bigger than the current largest dataset, MetaMathQA, and is the only one, except Lila, with a permissive license. Datasets marked with * have not been publicly released.

| Dataset | Size | Generating LM (Permissive License) |
|---|---|---|
| Lila [19] | 272K | - (✓) |
| MathInstruct [2] | 262K | GPT-4 (✗) |
| MetaMathQA [1] | 395K | GPT-3.5 (✗) |
| MathCodeInstruct [7] | 80K | GPT-4 + Self (✗) |
| WizardMath* [20] | 96K | GPT-3.5 (✗) |
| ToRA* [8] | 16K | GPT-4 (✗) |
| OpenMathInstruct-1 (Ours) | 1.8M | Mixtral (✓) |

## 2 Training Data Synthesis

### 2.1 Overview

**Setup.** Let $\mathcal{X} = \{(q_1, a_1), \cdots, (q_N, a_N)\}$ be a typical mathematical reasoning training dataset, where $q_i$ and $a_i$ denote the $i^{\text{th}}$ question and answer respectively. Optionally, the training dataset may include reference text solution $t_i$, which illustrates a trajectory from $q_i$ to $a_i$ using mathematical principles.[5] Besides the data, we assume access to a foundation LLM like `Mixtral-base`. The goal is to generate diverse, high-quality solutions for the training set problems using the LLM: a popular recipe for reasoning tasks [21, 22]. Recent work has also attempted augmenting training set problems [1, 2], but we limit our exploration to solution synthesis for existing problems in the benchmark.

**Solution Format.** We use the code-interpreter format for the synthesized solutions (see Figure 5 in Appendix for a sample solution). The code-interpreter format interweaves natural language reasoning with Python code blocks. It thus combines the computation precision of coding environments with the expressiveness of natural language reasoning, which is particularly suitable for mathematical reasoning tasks [8, 23]. To demarcate the start and end of a code block, we use the strings $\langle$llm-code$\rangle$ and $\langle$/llm-code$\rangle$. A code block is followed by its execution block, which is demarcated by $\langle$llm-code-output$\rangle$ and $\langle$/llm-code-output$\rangle$. During inference, the model invokes the Python interpreter to run the preceding code block after generating $\langle$/llm-code$\rangle$, appends the execution result in between the $\langle$llm-code-output$\rangle$ separators, and resumes the autoregressive model inference.[6]

**Approach.** We use few-shot prompting to synthesize solutions for the training sets of GSM8K and MATH. Formally, the prompt has the form:

$$\mathcal{I}\,(q_1, c_1), \cdots, (q_K, c_K)\,q'$$

where $\mathcal{I}$ represents a text-based instruction for the task, $\{q_1, \cdots, q_K\}$ represent $K$ problems representative of the dataset, $\{c_1, \cdots, c_K\}$ represent their respective solutions in the code-interpreter format, and $q'$ represents a question from the training set. Given this prompt, the base LLM generates a candidate solution $c'$ for the question $q'$. If $c'$ leads to the correct answer for the question $q'$, we add the pair $(q', c')$ to our fine-tuning set. For all our experiments, we choose $K = 5$, and the representative problems are chosen from the training set of the corresponding benchmark. In the instruction $\mathcal{I}$, we instruct the model to output the answer inside the \boxed{} block. The complete instruction is in Table 13 in Appendix B.4.

**Sampling Details.** We sample solutions with temperature=1.0 and top_p=0.95. We use the following constraints in our generation pipeline: (a) the total number of input-output tokens is limited to 4096, (b) a maximum of 512 new tokens after each code block, (c) a maximum of 3 code blocks, and (d) the generation halts after any code execution error. We use the TensorRT-LLM toolkit.[7]

---

[5] Both GSM8K and MATH have these text solutions.

[6] During training, we don't mask the code execution output surrounded by $\langle$llm-code-output$\rangle$ separators.

[7] https://github.com/NVIDIA/TensorRT-LLM

Table 2: Statistics of *unique* solutions generated by prompts described in Section 2.2. `Default` prompt refers to the single prompt used for the two benchmarks, `Mask-Text` refers to prompting the model with masked text solution, and `Subj` refers to prompting with subject-specific prompts (applicable only to MATH). Coverage % refers to the percentage of problems in the training set for which there's at least one solution among the generated solutions.

| Prompt | MATH | | | GSM8K | | |
| | # Samples | # Unique Solns. | Coverage (in %) | # Samples | # Unique Solns. | Coverage (in %) |
|---|---|---|---|---|---|---|
| Default | 224 | 177K | 80.1 | 128 | 434K | 99.1 |
| + Subj | 224 | 191K | 80.1 | - | - | - |
| Mask-Text | 224 | 192K | 85.9 | 128 | 602K | 99.9 |
| + Subj | 224 | 227K | 87.5 | - | - | - |
| Total | 896 | 787K | 93.0 | 256 | 1036K | 99.9 |

## 2.2 Prompting

In the previous section, we described our solution generation pipeline. A key ingredient of this pipeline is the few-shot prompt examples. We next describe the different prompting strategies explored in this work.

### 2.2.1 Default

We choose five representative examples of GSM8K and MATH to create the few-shot prompt for the respective datasets. For GSM8K, we use a mix of problems that require vanilla Python code and problems that are best solved using Python's *sympy* library. For MATH, we compose a 5-shot prompt with examples from different subjects. To reflect this diversity of reasoning paths required for MATH, we choose a mix of problems that require code-based solutions, text-based solutions, and a combination of both. The prompts used for the two datasets are presented in Appendix B.6.

For GSM8K, we sample 128 solutions per training problem, which gets a training set coverage of 99.1%. For MATH, we sample 224 solutions per training problem, which only achieves a training set coverage of 80.1%. This difference in coverage reflects the difficulty of the MATH benchmark compared to GSM8K, which has been noted in previous work as well [8, 9].

### 2.2.2 Subject-specific Prompting (Subj)

*Could the diversity of mathematical topics in MATH be a reason for the low training set coverage with a single 5-shot prompt?* To answer this question, we create subject-specific prompts for the seven subjects in the MATH benchmark, namely `algebra`, `geometry`, `intermediate algebra`, `number theory`, `prealgebra`, `precalculus`, and `probability` (See Table 11 in the appendix for the subject-wise split of MATH training data). The MATH benchmark also labels problems by their hardness level, with levels ranging from 1 to 5, where level 5 is the hardest. For creating subject-specific 5-shot prompts, we choose one example from each level for a given subject. For each of the seven prompts, we sample 32 solutions per problem and combine the data generated with all the prompts, which is equivalent to 32 x 7 = 224 solutions per problem. However, even with this fine-grained prompting, we only achieve a negligible gain in the training set coverage, though the total number of correct solutions increases by 14K (177K $\rightarrow$ 191K, see Table 2).

Combining this fine-tuning dataset with the earlier single `default` prompt dataset yields a training coverage of 85.1% for MATH, a boost of 5% absolute. However, achieving this coverage required sampling almost 450 solutions per problem (224 + 224 = 448). *Can we make the solution generation pipeline more efficient?*

### 2.2.3 Masked Text Solution Prompting (Mask-Text)

GSM8K and MATH benchmarks come with reference text solutions. Using these text solutions can, in theory, reduce the problem of code-interpreter solution generation to a translation problem from text to code. We initially experimented by prompting the LLM with:

$$\mathcal{I}\,(q_1, t_1, c_1), \cdots, (q_K, t_K, c_K)\, q', t'$$



**Masked Text Solution**

**Question**
Lynne bought 7 books about cats and 2 books about the solar system. She also bought 3 magazines. Each book cost $7 and each magazine cost $4. How much did Lynne spend in all?

**Ground-Truth Text Solution**
Lynne bought a total of 7 + 2 = 9 books. The books cost Lynne 9 x 7 = $63. For 3 magazines, Lynne spent 3 x 4 = $12. In total, Lynne spent 63 + 12 = $75

**Masked Text Solution**
Lynne bought a total of 7 + 2 = M books. The books cost Lynne M x 7 = N. For 3 magazines, Lynne spent 3 x 4 = P. In total, Lynne spent N + P = Q



Figure 2: A sample masked solution from GSM8K training set. The masked text solution only masks the intermediate computations, such as $9 \rightarrow M$ and $63 \rightarrow N$, and doesn't mask the amounts introduced in the question, such as 7, 2, and $4.

where $t_i$'s represent the text solution of representative problem $q_i$'s and $t'$ represents the text solution of the problem $q'$. Using the text solution in the prompt leads to a considerable increase in training set coverage. However, our manual analysis revealed that many solutions were *shortcuts*. E.g., trivial solutions such as `print(ANSWER)` or `The answer is ANSWER` where the `ANSWER` is copied from the text solution $t'$ in the prompt. Our attempts to filter out these trivial solutions proved challenging as there are many creative ways in which the generated solutions were cheating (see Figure 11 in Appendix).

To deter the possibility of such *shortcut* solutions where the results of intermediate computations or the final answer from the text solution are copied, we propose prompting with a *masked text solution*. Such solutions have all numbers in intermediate computations replaced with symbols. A sample masked text solution is shown in Figure 2. These masked text solutions are generated using few-shot prompting as follows:

$$\mathcal{I}_{\text{mask}} \ (q_1, t_1, t_1^{\text{mask}}), \cdots, (q_K, t_K, t_K^{\text{mask}}) \ q', t'$$

where $\mathcal{I}_{\text{mask}}$ represents the instruction for the solution masking task, and $\{t_1^{\text{mask}}, \cdots, t_K^{\text{mask}}\}$ represent masked text solutions corresponding to $\{t_1, \cdots, t_K\}$. For a detailed overview of the masked text solution generation pipeline, we refer the reader to Appendix B.5. Using these masked text solutions in the prompts significantly boosts the training set coverage for MATH, increasing from 80.1% $\rightarrow$ 85.9% for the single *default* prompt, and 80.1% $\rightarrow$ 87.5% for the subject-specific prompts. For GSM8K, it leads to the coverage increasing from 99.1% to 99.9%.

Table 2 summarizes the statistics of the solutions dataset generated via different prompts. The OpenMathInstruct-1 dataset is obtained by merging and deduplicating the problem-solution pairs resulting from the above-described prompt strategies. OpenMathInstruct-1 consists of 787K unique solutions for 6978 problems (out of 7500) in MATH and 1.04M unique solutions for 7469 problems (out of 7473) in GSM8K. To get to this final dataset, we also perform a few post-processing steps, which are described next.

## 2.3 Post-processing

The generated solutions can sometimes be *syntactically noisy* even if they lead to the right answer. We fix or remove the following solutions:

- Remove solutions with multiple `\boxed{}` blocks.
- Remove solutions with the ⟨llm-code⟩ string but not the ⟨/llm-code⟩ string.
- Remove text beyond the solution line with the answer, i.e., the `\boxed{}` block. See Figure 12 in the Appendix for an example solution where we perform trimming.

While these post-processing steps can fix some of the syntactic errors, filtering *semantically noisy*, i.e., solutions that get to the right answer with flawed reasoning [12], is a much harder problem and beyond the scope of this work. Anecdotally, we find such solutions to be rare in our corpus. See Figure 13 in the Appendix for a sample *semantically noisy* solution.

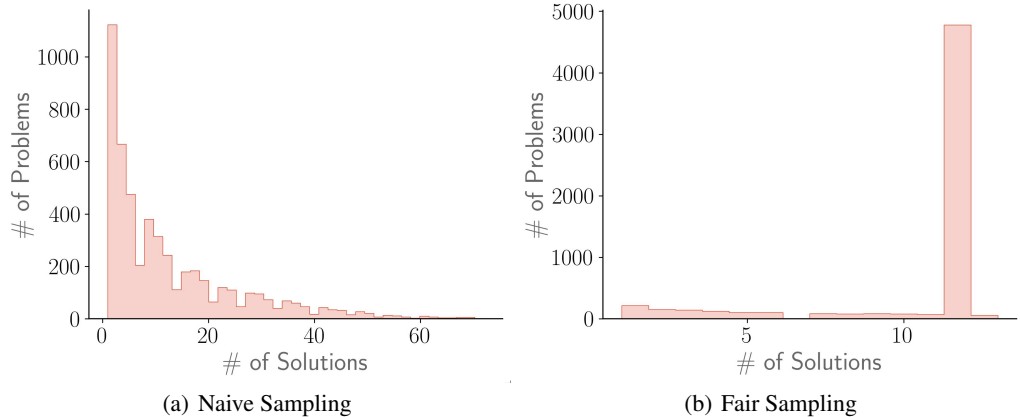

|  (a) Naive Sampling | (b) Fair Sampling |

Figure 3: Histogram of the number of solutions for problems in a 64K downsampled subset of MATH instances in OpenMathInstruct-1.

## 2.4 Data Selection

OpenMathInstruct-1 on average has hundreds of solutions per problem. These solutions can have different formats (code vs. text), and problems can have very different numbers of solutions in the dataset. Careful data selection allows for reduced training times and can also benefit performance. We detail the data selection strategies explored in this work.

### 2.4.1 Fair vs. Naive Downsampling

For a dataset like MATH, where problems have divergent difficulty levels, our solution generation strategy leads to a corpus where *easier* problems have a lot of solutions and *harder* problems have very few solutions (see Appendix A.2 for a detailed discussion on solution count). A *naive* strategy for downsampling treats every instance, i.e., problem-solution pair, as an equal. This problem-agnostic sampling perpetuates the imbalance of the original corpus, as seen in Figure 3(a). We propose a *fair* sampling alternate in which we iterate over all the problems round-robin and sample without replacement from the remaining solutions for each problem. This problem-dependent sampling ensures a more balanced representation of each problem in the downsampled dataset (see Figure 3(b)). Experimental results show that *fair* downsampling outperforms *naive* downsampling (Section 4.1.1).

### 2.4.2 Code-Preferred Solutions

The code-interpreter format allows for mixing code and text, and also text-based solutions without any code blocks. For GSM8K, the proportion of text-based solutions is 2%, but for MATH, their representation is 35.1%.[8] While natural language reasoning is more expressive, it lacks the precision of code-based solutions [24]. Suppose for a problem $q$ there are a total of $N_{total}$ correct solutions in the corpus, out of which $N_{code}$ represents the number of code-based solutions, and $N_{text}$ represents the text-based solutions. We propose the following two *code-preferential* data selection strategies:

- *Majority-Code*: If $N_{code} > N_{text}$, remove all the text-based solutions.
- *Any-Code*: If $N_{code} > 0$, remove all the text-based solutions.

Ablation experiments over the MATH subset of OpenMathInstruct-1 show the benefit of *code-preferential* data selection (Section 4.1.4).

## 3 Experimental Setup

### 3.1 Training Details

For all our experiments, including ablations, models of size 34B or smaller are trained for four epochs. A global batch size of 128 is used along with the AdamW optimizer with a weight decay of 1e-

---

[8]We detect the presence of code by searching for ⟨`llm-code`⟩ in the solution string.

Table 3: Comparison of our *OpenMath-finetuned* models with their *gpt-distilled* counterparts. We present results on popular mathematical reasoning tasks, namely, GSM8K, MATH, GSM-Hard, SVAMP, TabMWP, ASDiv, and MAWPS. For ToRA and MAmmoTH, we report the results of their "-Code(r)" versions whenever available since they are always better than their non-code counterparts. SC (k=50) denotes self-consistency decoding with 50 samples. We highlight the following results for a parameter range: best with SC, **best** and second best with greedy decoding.

| Size | Base Model | Model | GSM8K | MATH | GSM-Hard | SVAMP | TabMWP | ASDiv | MAWPS |
|------|------------|-------|-------|------|----------|-------|--------|-------|-------|
| - | GPT-4 (Code Interpreter) | | 97.0 | 69.7 | 77.6 | 94.8 | 95.9 | 92.6 | 97.7 |
| 7B | Llama-2 | WizardMath | 54.9 | 10.7 | - | 36.1 | - | - | - |
| | | MetaMath | 66.4 | 19.4 | - | | | | |
| | CodeLlama | MAmmoTH | 59.4 | 33.4 | - | 71.4 | - | - | - |
| | | ToRA | 72.6 | **44.6** | 56.0 | 70.4 | 51.6 | 78.7 | 91.3 |
| | | + SC (k=50) | 76.8 | 52.5 | - | - | - | - | - |
| | | OpenMath-CodeLlama | 75.9 | 43.6 | 60.1 | 79.6 | 56.0 | 77.7 | 93.5 |
| | | + SC (k=50) | 84.8 | 55.6 | - | - | - | - | - |
| | Mistral | MetaMath-Mistral-7B | 77.7 | 28.2 | - | - | - | - | - |
| | | MAmmoTH-7B-Mistral | 75.0 | 40.0 | - | - | - | - | - |
| | | WizardMath | **83.2** | 33.0 | - | - | - | - | - |
| | | OpenMath-Mistral-7B | 80.2 | 44.5 | **63.7** | **82.4** | **70.0** | **82.7** | **95.4** |
| | | + SC (k=50) | 86.9 | 57.2 | - | - | - | - | - |
| 13B | Llama-2 | WizardMath | 63.9 | 14.0 | - | 51.9 | - | - | - |
| | | MetaMath | 72.3 | 22.4 | - | - | - | - | - |
| | CodeLlama | MAmmoTH | 64.7 | 36.3 | - | 73.7 | - | - | - |
| | | ToRA | 75.8 | **48.1** | 60.5 | 75.7 | **65.4** | **81.4** | 92.5 |
| | | + SC (k=50) | 80.4 | 55.1 | - | - | - | - | - |
| | | OpenMath-CodeLlama | **78.8** | 45.5 | **61.9** | **78.8** | 59.7 | 81.2 | **93.6** |
| | | + SC (k=50) | 86.8 | 57.6 | - | - | - | - | - |
| 34B | CodeLlama | MAmmoTH | 72.7 | 43.6 | - | **84.3** | - | - | - |
| | | ToRA | **80.7** | **51.0** | 63.7 | 80.5 | **70.5** | **84.2** | 93.3 |
| | | + SC (k=50) | 85.1 | 60.0 | - | - | - | - | - |
| | | OpenMath-CodeLlama | **80.7** | 48.3 | **64.0** | 83.6 | 66.0 | 82.7 | **94.9** |
| | | + SC (k=50) | 88.0 | 60.2 | - | - | - | - | - |
| 70B | Llama-2 | WizardMath | 81.6 | 22.7 | - | 71.8 | - | - | - |
| | | MetaMath | 82.3 | 26.6 | - | - | - | - | - |
| | | MAmmoTH | 76.9 | 41.8 | - | 82.4 | - | - | - |
| | | ToRA | 84.3 | 49.7 | **67.2** | 82.7 | 74.0 | **86.8** | 93.8 |
| | | + SC (k=50) | 88.3 | 56.9 | - | - | - | - | - |
| | | OpenMath-Llama2 | **84.7** | 46.3 | 65.7 | 85.0 | 70.8 | 84.3 | 95.6 |
| | | + SC (k=50) | 90.1 | 58.3 | - | - | - | - | - |
| | CodeLlama | OpenMath-CodeLlama | 84.6 | **50.7** | 66.6 | **87.8** | **74.2** | 84.7 | **95.7** |
| | | + SC (k=50) | 90.8 | 60.4 | - | - | - | - | - |

2 [25] and dropout [26] of 0.1. We save one checkpoint per epoch for ablation experiments and two checkpoints per epoch for final model runs. The final checkpoint is created by averaging all the saved checkpoints. All experiments are performed using the NeMo toolkit[9] [27]. For the full set of training hyperparameters, see Appendix B.1.

## 3.2  Evaluation Setup

We evaluate our models on popular math reasoning benchmarks, namely GSM8K, MATH, GSM-Hard [24], SVAMP [28], TabMWP [29], ASDiv [30], and MAWPS [31]. For ablation experiments and hyperparameter selection, we create a validation set of 1K examples from the training set of GSM8K and MATH since both datasets lack an actual validation set. All the fine-tuned models are evaluated in the zero-shot setting. We use greedy decoding and self-consistency/majority voting [32] for evaluation. For majority voting, we found that using a lower temperature of 0.7 is beneficial compared to the data generation setup. We also deviate from the data generation setup by allowing the model to continue answer generation after code execution errors.

---

[9]https://github.com/NVIDIA/NeMo

# 4 Results

We finetune all the models on a mixture of (a) 512K fair downsampled GSM8K instances, and (b) 512K MATH instances with *any-code* filtering (Section 2.4).[10] Thus, the total finetuning corpus size is roughly 1.02M. We justify the data selection choices later in the ablation experiments.

Table 3 compares the performance of *OpenMath-finetuned* models against their *GPT-distilled* counterparts. Among the 7B models, our OpenMath-Mistral-7B is competitive with all the *GPT-distilled* models. It is second-best to WizardMath on GSM8K, and bested by ToRA by 0.1% on MATH.[11] Our models easily outperform both MetaMath [1] and MAmmoTH [2], even when controlling for the base fine-tuned model. Since WizardMath and ToRA finetuning datasets are not publicly available yet, OpenMathInstruct-1 presents a superior alternative to the publicly available MetaMathQA and MathInstruct datasets, which are used to fine-tune MetaMath and MAmmoTH, respectively.

With the increase in model parameters, our models continue to outperform MetaMath and MAmmoTH substantially. Compared to ToRA, with greedy decoding, we see a meaningful drop in performance on MATH, though our models are equal or better on GSM8K. With self-consistency (SC) decoding, however, our models outperform ToRA on both MATH and GSM8K. The substantial gains with SC can be attributed to the diversity of our fine-tuning data.

## 4.1 Ablations

We perform ablation experiments with the Mistral-7B as the base model. We report results on the 1K-sized validation subsets for MATH and GSM8K created by us.

Table 4: Comparison of performance of Fair vs Naive downsampling on our validation subset of GSM8K and MATH.

| Sampling | GSM8K | MATH |
|----------|-------|------|
| Naive | 74.3 | 35.0 |
| Fair | **75.3** | **37.0** |

Table 5: Comparison of Default vs Masked prompting on our validation subset of GSM8K and MATH.

| Prompt | GSM8K | MATH |
|--------|-------|------|
| Default | 73.8 | 36.9 |
| Masked | **77.7** | **37.4** |

### 4.1.1 Fair vs. Naive Downsampling

We finetune the base model on a dataset of 128K instances created by combining 64K naive or fair downsampled instances from the GSM8K and MATH portion of the data. Table 4 shows that the model fine-tuned on the data downsampled with fair sampling outperforms the one created by naive downsampling. The performance gap is particularly substantial for MATH, which suffers from a graver data imbalance than GSM8K in our corpus.

### 4.1.2 Default vs Masked Prompting

We finetune the base model on a dataset of 128K instances created by combining 64K fair-sampled instances from the GSM8K and MATH portion of the data generated using default and masked prompting. Table 5 shows that the model fine-tuned on the data generated using masked prompting outperforms the one created by default prompting on both GSM8K and MATH. Thus, the gains in the training set coverage with masked prompting (Section 2.2.3) also translate to finetuning performance.

### 4.1.3 Impact of Fine-Tuning Dataset Size

To determine the impact of the size of the fine-tuning dataset, we create datasets of size 128K/256K/512K by combining 64K/128K/256K fair downsampled equally-sized subsets of GSM8K and MATH. Table 6 shows that the performance increases for both GSM8K and MATH with the increase in the fine-tuning dataset size. We didn't find any benefit from training the models for more steps, so the performance gain is attributable to the increased data size.

---

[10]The actual number of MATH instances is 511,677.

[11]Our grading script scores the publicly released ToRA outputs about 2-3% lower than the reported numbers. We believe that ToRA uses some heuristics to extract answers when the model doesn't generate answers in the correct format.

Table 6: Effect of fine-tuning dataset size on performance on our validation subset of GSM8K and MATH.

| Dataset Size | GSM8K | MATH |
|:---:|:---:|:---:|
| 128K | 75.3 | 37.0 |
| 256K | 79.0 | 38.6 |
| 512K | 81.0 | 41.6 |

Table 7: Comparison of default vs subject-wise prompt performance on our MATH validation subset.

| Prompt | Pass@1 | SC (k=4) |
|:---|:---:|:---:|
| Default | 39.1 | 41.7 |
| Subject | 38.3 | 44.5 |

Table 8: Impact of code-preferential data selection on our MATH validation subset performance.

| Prompt | Pass@1 | SC (k=4) |
|:---|:---:|:---:|
| Default | 37.4 | 45.2 |
| Majority-Code | 39.8 | 42.6 |
| Any-Code | 39.4 | 42.6 |

### 4.1.4 MATH-only Ablations

This section presents the ablation results for only the MATH portion of OpenMathInstruct-1. We finetune the base model on a 128K *fair* downsampled subset to control for data size.

**Default vs Subject-Specific Prompting.** In section 2.2.2, we motivated using subject-specific prompts, which ultimately didn't result in much training set coverage difference. *But how are the solutions generated by the combination of subject-wise prompts different from a single default prompt?* To answer this, we create a subset of 128K instances generated with the default prompt/subject-specific prompts. Table 7 compares the finetuning performance on these two splits on our MATH validation subset. While the model trained on the *subject-specific* subset underperforms the model trained on the *default* subset with greedy decoding, the trend is decisively reversed for self-consistency decoding with four samples. This suggests that the subset collected with subject-specific prompts has a higher diversity of solutions than the ones collected using a single prompt.

**Code-Preferential Subsets.** In this ablation, we determine the impact of code-preferential solution selection strategies proposed in Section 2.4.2. Table 8 shows that code-preferential solution strategies aid the greedy decoding performance. However, the reduction in solution diversity arguably results in a performance drop with self-consistency decoding (text-based solutions are about one-third of the original corpus). Based on these results and because *Any-Code* results in a smaller finetuning dataset (512K compared to 664K with *Majority-Code*), we chose to use the *Any-Code* subset.

## 5 Analysis

We analyze the performance of the ablation model trained on 512K instances from Section 4.1.3. We limit the discussion to the MATH benchmark, where the model scores 41.6% on our validation subset.

**Performance-split by Subjects and Levels.** Figure 4 presents the performance split by subjects and levels on the MATH validation subset. Among subjects, we see that the model's worst performance is on geometry, which can be attributed to the lack of multi-modality in our base models [23]. We see a monotonic decrease in performance with the increase in hardness level which is to be expected [23]. The model scores 72.4% on Level 1 problems and only 16.3% on the hardest problems, i.e., Level 5.

**Error Analysis.** Table 9 shows that the model performs an absolute 13.3% better when using code for answering questions in comparison to when not using it. We find that some of the errors made by text-based solution could have been avoided by preferring code-based solution; see Figure 16 for a sample solution where the model makes an arithmetic calculation error. This analysis provides another support for our proposal and use of code-preferred solutions from Section 2.4.2.

Table 10 presents the count of different error categories. For code-based solutions, we find that almost 74% of the errors in such solutions are due to reasoning errors, and the remaining 26% are attributable to execution-related issues. We present sample solutions from these error categories in Appendix B.3.

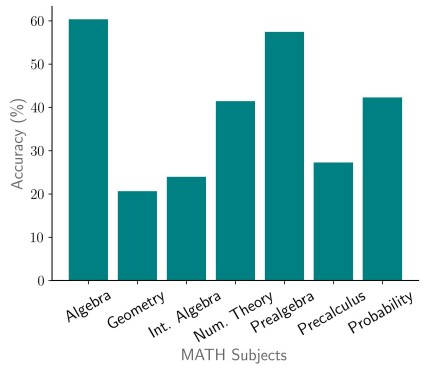

(a) Subject-wise performance

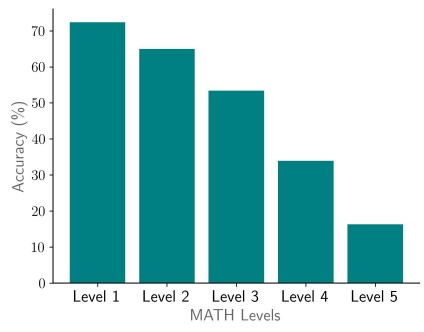

(b) Level-wise performance

Figure 4: Performance split by subjects and levels on our MATH validation subset.

Table 9: Performance split based on solution format. Solutions without ⟨llm-code-output⟩ string are considered to be *text-based*.

| Solution Type | Accuracy (in %) | Count |
|---|---|---|
| Text-based | 32.0 | 278 |
| Code + Text | 45.3 | 722 |
| Total | 41.6 | 1000 |

Table 10: Types of errors and their counts.

| Error Type | Count |
|---|---|
| Text Reasoning Error | 189 |
| Code Reasoning Error | 292 |
| Code Execution Error | 78 |
| Code timeout | 15 |
| Max code executions reached | 10 |
| Total | 584 |

## 6  Related Work

**Mathematical Reasoning and LLMs.**   Recently, a plethora of work has been done on enhancing the mathematical reasoning capabilities of LLMs.  Inference techniques such as Chain-of-Thought [33], its programmatic counterpart, Program of Thought [24, 34], Self-Consistency [32], and Self-Verification [23] have been shown to significantly improve the reasoning capabilities of LLMs.

Pretraining language models on math-heavy content has resulted in foundation LLMs such as Minerva [35], Galactica [36], Llemma [37], and DeepSeekMath [38] with stronger mathematical skills out-of-the-box. A more direct approach of dataset-specific training does *instruction fine-tuning* on problem-solution pairs derived from math reasoning datasets. Our work falls in this latter category and bears similarity with recent work such as RFT [39], ToRA [8], MAmmoTH [2], MetaMath [1] and MathCoder [7].  We differ from the previous work along one factor or a combination of the following factors: (a) reliance on GPT-3.5/4, (b) solution format, and (c) use of ground truth text solution in synthesizing code-based solutions.

**Knowledge Distillation via Synthetic Data.**   Recent work exploring the use of targeted *synthetic* data generated by large foundation models for pre-training/instruction tuning smaller LLMs has led to tremendous progress in skills of these smaller LLMs [5, 6, 4, 40, 41, 42].

## 7  Conclusion

We introduce OpenMathInstruct-1, a math instruction tuning dataset with 1.8M problem-solution pairs which is released with a commercially permissive license.  Compared to previous work, the OpenMathInstruct-1 dataset is at least four times bigger.  With our proposed prompting novelty of using *masked text solutions* and some brute-force scaling, we achieve training set coverage of 99.9% for the GSM8K benchmark and 93% for the challenging MATH benchmark. The quality of these synthesized solutions is illustrated by finetuning experiments, which show models achieving performance comparable to or better than their *gpt-distilled* counterparts. To support the open-source efforts in this direction, we publicly release all our fine-tuned models, code, and the OpenMathInstruct-1 along with a further 6.6M incorrect sampled solutions.

## Limitations and Potential Risks

Our work aims to improve the mathematical reasoning of open-source models using open-source models. In pursuit of this goal, we create a synthetic dataset, OpenMathInstruct-1, that our experiments show aids the performance on existing math benchmarks. Below, we list the key limitations of our work:

- Our manual analysis reveals solutions that get the right answer but via flawed reasoning (Figure 13 in Appendix). Removing these *semantically noisy* solutions is beyond the scope of the current work. This means a lack of guarantee about the quality of our synthetically generated solutions.

- Improving performance on in-domain math benchmarks may not translate to performance gain on other related tasks. The drop in performance on GSM-Hard compared to GSM indicates that our models may not be robust to input perturbations, though, they are at par with previous work.

While we don't foresee any material risk due to our work, using our imperfect dataset and models to perform tasks, such as evaluating student assignments or building a math tutor, carries risk.

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

# A   Data Statistics

## Sample Code-Interpreter Style Solution

**Question**

A department store displays a 20% discount on all fixtures. What will be the new price of a 25 cm high bedside lamp that was worth $120?

**Code-Interpreter Style Solution**

Let's solve this problem using Python code.

```
<llm-code>
discount_percent = 20
price_before_discount = 120
discount = discount_percent / 100
discount_amount = price_before_discount * discount
price = price_before_discount - discount_amount
price
</llm-code>
<llm-code-output>
96.0
</llm-code-output>
```

So the new price of the lamp is $\boxed{96}$ dollars.

Figure 5: Code-Interpreter style solution for a training set problem from GSM8K.

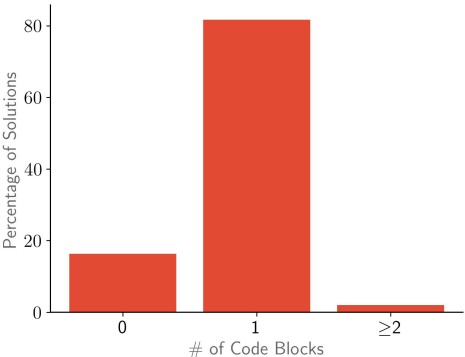

Figure 6: Distribution of solutions with different code-block counts in OpenMathInstruct-1.

### A.1 Code-Block Count Frequencies

We're using the code-interpreter solution format, which allows for flexible reasoning in text along with precision in code-based reasoning (see Figure 10). The code-interpreter allows solving a problem by breaking it into multiple code blocks. Most of the solutions in OpenMathInstruct-1 have 0 or 1 code blocks, 16.4% and 81.7% of the solutions, respectively (see Figure 6). The remaining 2% of the solutions have two or more code blocks. Figure 7 shows an example of a solution using two code blocks.

### A.2 Frequency of solutions per problem in OpenMathInstruct-1

Figure 8 shows the histogram of the frequency of the number of solutions in OpenMathInstruct-1 for problems in GSM8K and MATH. For GSM8K, 57.4% problems (4292 out of 7473) have more than half the sampled solutions, i.e., $> 128$ solutions out of 256 sampled solutions, that are *valid*. For MATH, the solution frequency histogram mimics a power-law curve, with most problems having very few solutions and very few problems with many solutions. Concretely, 19% problems (1324 out of

## Solution with Two Code Blocks

### Question

Let $d$ and $e$ denote the solutions of $2x^2 + 3x - 5 = 0$. What is the value of $(d-1)(e-1)$?

### Generated Solution

Let's solve the quadratic equation using Sympy:

```
<llm-code>
from sympy import Symbol, solve, Eq, simplify

# Define the variable x
x = Symbol('x')

# Define the equation
eq = 2*x**2 + 3*x - 5

# Solve the equation
roots = solve(eq, x)

# Print the solutions
print("The solutions are:")
for root in roots:
    print("x = ", simplify(root))
</llm-code>
<llm-code-output>
The solutions are:
x = -5/2
x = 1
</llm-code-output>
```

So $d = -5/2$ and $e = 1$.
Let's calculate the answer:

```
<llm-code>
d = -5/2
e = 1
d_minus_1 = d - 1
e_minus_1 = e - 1
result = d_minus_1 * e_minus_1
print("The value of (d-1)(e-1) is:", result)
</llm-code>
<llm-code-output>
The value of (d-1)(e-1) is: -0.0
</llm-code-output>
```

So the answer is $\boxed{-0.0}$.

Figure 7: Sample solution with multiple code blocks. The first code block computes the roots of the given quadratic equation and the second block computes the expression involving them.

6978) with any solutions in OpenMathInstruct-1 have $\leq 10$ valid solutions out of the 896 sampled solutions per problem. To further highlight the hardness of the MATH benchmark in comparison to GSM8K, just 3.1% problems (233 out of 7500) have more than half the sampled solutions ($> 448$ out of 896 sampled solutions), that are *valid*.

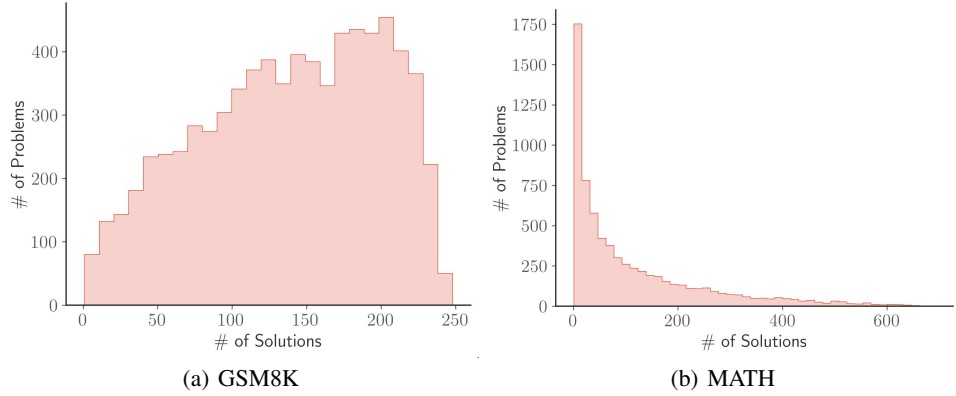

(a) GSM8K         (b) MATH

Figure 8: Histogram of the number of solutions for problems in GSM8K and MATH.

## Diversity of Solutions

### Question

Let $t(x) = \sqrt{3x+1}$ and $f(x) = 5 - t(x)$. What is $t(f(5))$?

### Solution 1

To find $t(f(5))$ we can substitute $x = 5$ into the functions $t(x)$ and $f(x)$ and then find the value of $t(f(5))$. To do the calculations we'll use sympy library.

```
<llm-code>
from sympy import symbols,
 sqrt, simplify

# Define the symbols
x, f = symbols('x f')

# Define the functions
t = sqrt(3*x + 1)
f = 5 - t

# Evaluate t at f(5)
t.subs(x, f.subs(x, 5))
</llm-code>
<llm-code-output>
2
</llm-code-output>
```

So the value of $t(f(5))$ is $\boxed{2}$.

### Solution 2

First let's calculate $t(x)$ for $x = 5$:
$t(5) = \sqrt{3*5+1} = \sqrt{16} = 4$.

Then let's calculate $f(x)$ for $x = 5$:
$f(5) = 5 - t(5) = 5 - 4 = 1$.

Finally let's calculate $t(f(5))$:
$t(f(5)) = t(1) = \sqrt{3*1+1} = \sqrt{4} = 2$.

So the answer is $\boxed{2}$.

### Solution 3

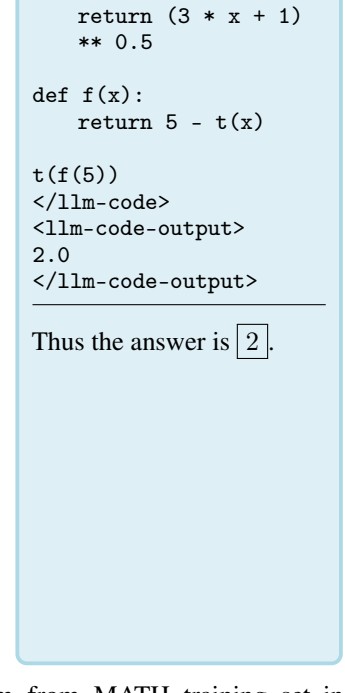

Thus the answer is $\boxed{2}$.

Figure 9: Three representative solutions for a sample problem from MATH training set in OpenMathInstruct-1. The first and third solutions use code, with the first one using `sympy` library. The second solution is text-based.

Table 11: MATH training set decomposition by subjects.

| Subject | # of Training Instances |
|---|---|
| Algebra | 1744 |
| Geometry | 870 |
| Intermediate Algebra | 1295 |
| Number Theory | 869 |
| Prealgebra | 1205 |
| Precalculus | 746 |
| Probability | 771 |
| Total | 7500 |

# B  Miscellaneous

## B.1  Training Hyperparameters

Table 12: Details of training hyperparameters for finetuning the different base models. LR=Learning rate, TP=Tensor Parallel, PP=Pipeline Parallel.

| | Epochs | LR | # of GPUs | TP | PP |
|---|---|---|---|---|---|
| Mistral-7B | 4 | 1e-6 | 64 | 4 | 1 |
| CodeLlama-7B | 4 | 2e-5 | 64 | 4 | 1 |
| CodeLlama-13B | 4 | 2e-5 | 64 | 4 | 1 |
| CodeLlama-34B | 4 | 1e-5 | 128 | 8 | 1 |
| Llama 2-70B | 2 | 1e-5 | 256 | 8 | 2 |
| CodeLlama-70B | 3 | 1e-5 | 256 | 8 | 2 |

Table 12 details the hyperparameters used for finetuning the different base models. Due to compute limitations, for 70B models, we limit the training to 2 epochs for Llama2 and 3 epochs for CodeLlama. For CodeLlama models, we use the learning rate of 2e-5 used by ToRA models [8]. For the Mistral-7B model, we use a lower learning rate of 1e-6 based on our hyperparameter search.

## B.2  Sample Solutions

In this section, we illustrate sample solutions representative of different phenomena encountered during the creation of OpenMathInstruct-1.

- Figure 9 presents a sample problem from the MATH training set with its representative solutions from the OpenMathInstruct-1 dataset.

- Figure 10 shows a sample solution that utilizes the strength of the code-interpreter solution format with reasoning in both code and natural language.

- Figure 11 demonstrates a sample solution generated when the reference solution is used in the few-shot prompt. The model copies the children's ages from the reference solution and initializes the `child_age` variable. Such solutions are the reason why we propose the use of masked text solutions in the prompt.

- Figure 12 illustrates a sample solution where the solution goes beyond answering the question, with the model generating coherent but unrelated text for the input problem.

- Figure 13 shows a sample solution where the generated solution gets the right answer but through flawed reasoning. These *semantically noisy* solutions are much harder to detect with simple syntactic filters. One solution might be to use models like GPT-4 to grade the generated solutions as done in recent work [5]. We leave the work of developing such semantic filters for future work.

| Code + Text Reasoning |
|---|

**Question**

Let $p(x) = \sqrt{-x}$, and $q(x) = 8x^2 + 10x - 3$. The domain of $p(q(x))$ can be written in the form $a \leq x \leq b$. Find $b - a$.

**Generated Solution**

Let's use sympy to solve for the domain of $p(q(x))$.

```
<llm-code>
import sympy as sp

# define the symbols
x = sp.symbols('x')

# define the functions
p = sp.sqrt(-x)
q = 8*x**2 + 10*x - 3

# solve for the domain of p(q(x))
domain = sp.solve(q >= 0, x)

# print the domain
print(domain)
</llm-code>
<llm-code-output>
((1/4 <= x) & (x < oo)) | ((-oo < x) & (x <= -3/2))
</llm-code-output>
```

So the domain is $x \in [1/4, \infty) \cup (-\infty, -3/2)$.

The difference between the upper and lower bounds is $b - a = \boxed{1/4 - (-3/2)}$.

Figure 10: Sample solution where the model performs the initial reasoning in code and performs the last reasoning steps to answer the question in natural language. This shows the strength of the code-interpreter solution format for mathematical reasoning.

## B.3 Error Analysis of Solutions Generated by Fine-tuned Model

In this section, we illustrate instances of the different kinds of errors made by the ablation model analyzed in Section 5.

- Figure 14 demonstrates a sample where the model performs correct reasoning while generating the code. However, the model falters at copying the code output and generates a new answer.

- Figure 15 presents an example where the model generates correct but inefficient code that doesn't finish in our stipulated execution time and results in the *timeout* error.

- Figure 16 illustrates a sample where the model performs correct reasoning but falters in arithmetic calculation (multiplication). Failure at arithmetic computation has been a known issue with LLMs and justifies our preference for code-based solutions.

- Figure 17 shows an example where the model generates a wrong code block to solve the problem and repeatedly generates the same code block till it runs into our maximum execution constraint. This example shows a limitation of our current dataset where we lack examples in which the model recovers from its error, unlike GPT-4 [23].

- Figure 18 shows a sample solution where the code generated in the solution runs into an execution error. Nevertheless, the model still generates an *incorrect* answer to the question.

**Question**

Jolene and Phil have four children, each with the same birthday. They gave birth to their first child exactly 15 years ago. They gave birth to their second child exactly one year after the birth of their first child. They gave birth to their third child on the fourth birthday of their second child. Two years after the birth of their third child, they gave birth to their fourth child. How old, in years, is their fourth child?

**Reference Solution**

Their 1st child was born 15 years ago, and therefore is 15 years old.
Their 2nd child was born 1 year after their 15-year-old child, and therefore is 15-1=14 years old.
Their 3rd child was born 4 years after their 14-year-old child, and therefore is 14-4=10 years old.
Their 4th child was born 2 years after their 10-year-old child, and therefore is 10-2=8 years old.

**Generated Solution**

Let's write down a python script to answer this problem.

```
<llm-code>
children = ['first', 'second', 'third', 'fourth']
child_age = [15, 14, 10, 8]
number_of_children = len(children)
children = children[:-1] # let's get rid of the youngest one since we already know that one
dictionary = dict(zip(children, child_age))
dictionary
</llm-code>
<llm-code-output>
{'first': 15, 'second': 14, 'third': 10}
</llm-code-output>
```

The answer is $\boxed{8}$ years old.

Figure 11: Sample *shortcut* solution generated from prompting the LLM with the reference solution in context. The code initializes the variable `child_age` with the children's ages copied from the reference solution. The generated code then performs ad-hoc computations before answering the question in the text.

## B.4 Instructions for Few-shot Data Generation

Table 13: Instructions for prompting the model.

| Task | Instruction |
|------|-------------|
| Few-shot prompt ($\mathcal{I}$) | Here are some examples of questions and solutions followed by a new question that you need to solve. Make sure to put the answer (and only answer) inside \boxed{}. |
| Few-shot prompt text masking ($\mathcal{I}_{mask}$) | Here are some examples of questions, solutions, and their masked solutions followed by a new question and solution that you need to mask. The goal is to ensure that the masked solution doesn't have any of the numerical values not mentioned in the question. So intermediate values calculated in the solution are to be masked by single letter capital variables, such as M, N. |
| Zero-shot prompt for fine-tuned models | System: You're an expert Python programmer and mathematician. Help the user to solve this problem using code when necessary. Make sure to put the answer (and only answer) inside \boxed{}. |

Table 13 details the instructions used for the different generation tasks.

## Solution Requiring Trimming

**Question**

Caroline can make eleven lassis out of two mangoes. How many lassis can she make out of twelve mangoes?

**Generated Solution**

Let $f(x)$ be the number of lassis she can make out of $x$ mangoes. From the question, we can see that $f(2) = 11$. Using basic algebra, we can see that $f(12) = \boxed{66}$.

Let's verify this with sympy.

```
<llm-code>
import sympy as sp

# define the unknown function
x, y = sp.symbols('x y')

# let's define the parabola
parabola = sp.Eq(y, x**2 + b*x + c)

# substitute points into parabola equation and solve for b, c
point_1 = parabola.subs({x: -1, y: -11})
point_2 = parabola.subs({x: 3, y: 17})
solutions = sp.solve((point_1,point_2), (b, c))
solutions[b]
</llm-code>
```

**Trimmed Solution**

Let $f(x)$ be the number of lassis she can make out of $x$ mangoes. From the question, we can see that $f(2) = 11$. Using basic algebra, we can see that $f(12) = \boxed{66}$.

Figure 12: Sample solution where the model generates unrelated text after answering the question. During post-processing, we trim the solution by removing text beyond the solution line with the answer.

### B.5 Masked Text Solution Generation

We generate masked text solutions using a pipeline very similar to the solution generation pipeline. We use the following procedure:

- Generate eight candidate masked solutions.
- Filter out solutions that have very different lengths.
- Filter out solutions that have the final answer. This ensures that our masked solutions are at least masking the final answer.
- Rank solutions where higher-ranked solutions have fewer numbers. This is essentially a heuristic to ensure we pick solutions where the intermediate computation is masked out the most.

### B.6 Few-shot prompts used for GSM8K and MATH

**Question**

The areas of two squares are in the ratio $25 : 36$. What is the ratio of their perimeters? Express your answer in the form $a : b$.

**Reference Solution**

In general, if the ratio of the perimeters of two similar figures is $a/b$, then the ratio of their areas is $(a/b)^2$. So in this case, $(a/b)^2 = 25/36$, or $a/b = 5/6$.

So the answer is $\boxed{5 : 6}$.

**Generated Solution**

Let's use sympy to print out the difference between the ratio of their perimeters and each of the options.

```
<llm-code>
from sympy import Rational, Abs

# areas are in the ratio 25:36
area_ratio = Rational(25, 36)

# list of options
options = [Rational(5, 6), Rational(5, 4), Rational(5, 3), Rational(5, 2),
Rational(5, 1)]

# let's print out the differences
[Abs(area_ratio - frac_option) for frac_option in options]
</llm-code>
<llm-code-output>
[5/36, 5/9, 35/36, 65/36, 155/36]
</llm-code-output>
```

Let's now check which difference is the smallest.

```
<llm-code>
import numpy as np

# Calculate the idx of the closest option
min_idx = np.argmin([5/36, 5/9, 35/36, 65/36, 155/36])

# Print the closest option
print(options[min_idx])
</llm-code>
<llm-code-output>
5/6
</llm-code-output>
```

So the answer is $\boxed{5 : 6}$.

Figure 13: Sample solution where the model uses completely flawed reasoning to arrive at the correct answer. Such flawed reasoning is hard to detect, but fortunately, these solutions are rare.

**Question**

What is the 100th digit to the right of the decimal point in the decimal representation of $\frac{13}{90}$?

**Generated Solution**

We can use sympy to calculate the decimal representation of $\frac{13}{90}$ and then extract the 100th digit.

```
<llm-code>
from sympy import Rational, N

# Calculate the decimal representation of 13/90
decimal_rep = N(Rational(13, 90), 100)

# Extract the 100th digit
digit = int(str(decimal_rep)[-1])

print(digit)
</llm-code>
<llm-code-output>
4
</llm-code-output>
```

So the 100th digit to the right of the decimal point in the decimal representation of $\frac{13}{90}$ is $\boxed{7}$.

Figure 14: The model performs correct reasoning till code generation. But rather than copying the code output (4) it concludes with a different answer (7).

**Question**

Let $a_1, a_2, \ldots$ be a sequence for which $a_1 = 2$, $a_2 = 3$, and $a_n = \frac{a_{n-1}}{a_{n-2}}$ for each positive integer $n \geq 3$. What is $a_{2006}$?

**Generated Solution**

Let's write a function that calculates $a_n$ for a given $n$.

```
<llm-code>
def a_n(n):
    if n == 1:
        return 2
    elif n == 2:
        return 3
    else:
        return a_n(n-1) / a_n(n-2)

print(a_n(2006))
</llm-code>
<llm-code-output>
None
</llm-code-output>
```

So $a_{2006} = \boxed{1/5}$.

Figure 15: The model generates recursion-based code to solve the problem. While the code is technically correct, the lack of memoization means the code ends up being extremely inefficient and runs into timeout.

Calculation Error

**Question**

Our basketball team has 12 members, each of whom can play any position. In how many ways can we choose a starting lineup consisting of a center, a power forward, a shooting forward, a point guard, and a shooting guard?

**Generated Solution**

We can choose a starting lineup in $12 \times 11 \times 10 \times 9 \times 8 = \boxed{11880}$ ways.

Figure 16: Sample solution where the model performs the right text reasoning but makes calculation mistake.

**Question**

In $\triangle ABC$, $AB = 425$, $BC = 450$, and $AC = 510$. An interior point $P$ is then drawn, and segments are drawn through $P$ parallel to the sides of the triangle. If these three segments are of an equal length $d$, find $d$.

**Generated Solution**

Let's use sympy to solve this problem.

```
<llm-code>
from sympy import symbols, Eq, solve

# Define the variables
AB, BC, AC, d = symbols('AB BC AC d')

# Define the equations
eq1 = Eq(AB, 425)
eq2 = Eq(BC, 450)
eq3 = Eq(AC, 510)
eq4 = Eq(AB + BC, 2 * d)
eq5 = Eq(BC + AC, 2 * d)
eq6 = Eq(AC + AB, 2 * d)

# Solve the equations
solutions = solve((eq1, eq2, eq3, eq4, eq5, eq6), (AB, BC, AC, d))
solutions
</llm-code>
<llm-code-output>
[]
</llm-code-output>
```

The solutions are empty, so we can't solve this problem using sympy.
Let's try to solve it manually.
We can see that the sum of any two sides of a triangle is greater than the third side.
So we can write the following inequalities:
$AB + BC > AC$
$BC + AC > AB$
$AC + AB > BC$

Let's rewrite them using the given values:
$425 + 450 > 510$
$450 + 510 > 425$
$510 + 425 > 450$

We can solve these inequalities using sympy:
$\cdots$

Figure 17: The model generates the first code block, which is logically incorrect and results in an empty output. The following text block identifies the problem with the output but performs incorrect reasoning leading up to the generation of the second code block. The model fails to recover from these errors and ends up generating code-text blocks that are verbatim copies of the first code-text block. The generation stops at the sixth code block, which is our limit on the number of code-block executions during inference.

## Code Execution Error

**Question**

The area of a triangle is 600 square feet. Find the altitude, in feet, of the triangle if the length of the corresponding base is 30 feet.

**Generated Solution**

Let's use sympy to solve this problem.

```
<llm-code>
from sympy import symbols, Eq, solve

# define the variables
base, altitude = symbols('base altitude')

# area of the triangle
area_eq = Eq(base * altitude / 2, 600)

# length of the corresponding base
base_eq = Eq(base, 30)

# solve the equations
solutions = solve((area_eq, base_eq), (base, altitude))

# print the altitude
print(solutions[altitude])
</llm-code>
<llm-code-output>
[0;31m--------------------------[0m
[0;31mTypeError[0m Traceback (most recent call last)
File [0;32m<ipython-input-1-f95732badac7>:16[0m
[1;32m 13[0m solutions [38;5;241m=[39m solve((area_eq, base_eq), (base, altitude)
)
[1;32m 15[0m [38;5;66;03m# print the altitude[39;00m
[0;32m---> 16[0m [38;5;28mprint[39m([43msolutions[49m[43m[[49m[43maltitude[49m[43m][49m)

[0;31mTypeError[0m: list indices must be integers or slices, not Symbol
</llm-code-output>
```

So the altitude is $\boxed{20}$.

Figure 18: Sample solution where the model generated code runs into execution error. Nevertheless, the model still generates an *incorrect* answer to the question.

Here are some examples of questions and solutions followed by a new question
 that you need to solve.
Make sure to put the answer (and only answer) inside \boxed{}.

Question:
Missy had a giant piggy bank in her bedroom. Every day she would search the
house for change to put in her bank. After 4 years, the bank was opened and
it contained $450 in change. If the second, third, and fourth-year she
doubled the amount of money she put in the bank from the amount she had put
in the previous year, how much money, in dollars, did she put in the bank
the first year?

My solution:
Missy started with some amount of money in the first year.
She was then doubling the amount of money every year for 4 years and ended
up with $450.
Let's write down an equation for this problem and solve it using sympy.
<llm-code>
from sympy import solve, symbols
first_year_money = symbols('first_year_money')
second_year_money = 2 * first_year_money
third_year_money = 2 * second_year_money
fourth_year_money = 2 * third_year_money
# Solve for first year money given that the combined saving is 450
result = solve(first_year_money + second_year_money + third_year_money +
fourth_year_money - 450, first_year_money)[0]
result
</llm-code>
<llm-code-output>
30
</llm-code-output>
Thus Missy put \boxed{30} dollars in the bank the first year.

Question:
Pete has to take a 10-minute walk down to the train station and then board a
 1hr 20-minute train to LA. When should he leave if he cannot get to LA
later than 0900 hours? (24-hr time)

My solution:
Since Pete needs to take a 10 minutes walk and then a 1 hour 20 minutes
train ride, he will spend a total of 1 hour and 30 minutes.
This means that he needs to leave 1 hour and 30 minutes earlier than 09:00
hours.
Subtracting 1 hour and 30 minutes from 09:00 hours we get \boxed{07:30}
hours.

Question:
Mark deposited $88 in a bank. Bryan deposited $40 less than five times as
much as Mark. How much did Bryan deposit in the bank?

My solution:
Let's solve this problem using Python code.
<llm-code>
mark_deposit = 88
five_times_mark_deposit = 5 * mark_deposit

```
bryan_deposit = five_times_mark_deposit - 40
bryan_deposit
</llm-code>
<llm-code-output>
400
</llm-code-output>
So Bryan deposited \boxed{400} dollars in the bank.
```

Question:
A department store displays a 20% discount on all fixtures. What will be the
 new price of a 25 cm high bedside lamp that was worth $120?

My solution:
Let's solve this problem using Python code.
```
<llm-code>
discount_percent = 20
price_before_discount = 120
discount_portion = discount_percent / 100
discount_amount = price_before_discount * discount_portion
price_after_discount = price_before_discount - discount_amount
price_after_discount
</llm-code>
<llm-code-output>
96.0
</llm-code-output>
```
So the new price of the lamp is \boxed{96} dollars.

Question:
James opens up a flower shop. He needs to pay rent of $1200 a week with an
additional 20% of rent to pay for utilities and he has 2 employees per shift
 with the store open 16 hours a day for 5 days a week. If he pays each
employee $12.50 an hour, what are his weekly expenses to run the store?

My solution:
The cost consists of rent, utilities, and employee salaries. Let's compute
each of them separately and then add them up.
```
<llm-code>
# rent cost
rent_per_week = 1200
# utility cost
utility_per_week = rent_per_week * 20 / 100
# employee cost
employee_work_hours = 16
work_days_per_week = 5
employee_work_hours_per_week = work_days_per_week * employee_work_hours
number_of_employees = 2
employee_cost_per_hour = 12.5
employees_cost_per_week = number_of_employees * employee_work_hours_per_week
 * employee_cost_per_hour
# add the three to get total cost
cost_per_week = rent_per_week + utility_per_week + employees_cost_per_week
cost_per_week
</llm-code>
<llm-code-output>
3440.0
</llm-code-output>
```
Thus James's weekly expences add up to \boxed{3440} dollars.

Question:
Tim has 30 less apples than Martha, and Harry has half as many apples as Tim. If Martha has 68 apples, how many apples does Harry have?

My solution:

Here are some examples of questions and solutions followed by a new question
 that you need to solve.
Make sure to put the answer (and only answer) inside \boxed{}.

Question:
A parabola with equation $y=x^2+bx+c$ passes through the points $(-1,-11)$
and $(3,17)$. What is $c$?

My solution:
Let's write down an equation for the parabola and solve for $c$ using sympy.
```
<llm-code>
import sympy as sp

# define the symbols
x, y, b, c = sp.symbols('x y b c')

# define the parabola equation
parabola_eq = sp.Eq(y, x**2 + b*x + c)

# the parabola passes through the points (-1,-11) and (3,17)
# so we substitute these points into the parabola equation
point_1 = parabola_eq.subs({x: -1, y: -11})
point_2 = parabola_eq.subs({x: 3, y: 17})

# we now have two equations and two unknowns (b and c)
# we will solve for b and c
solutions = sp.solve((point_1,point_2), (b, c))
solutions[c]
</llm-code>
<llm-code-output>
-7
</llm-code-output>
```
So c is \boxed{-7}

Question:
Let $f(x)$ be an odd function. Is $f(f(x))$ even, odd, or neither?

Enter "odd", "even", or "neither".

My solution:
The definition of an odd function is that $f(-x) = -f(x)$.
Applying this to $f(f(-x))$ we get $f(f(-x)) = f(-f(x)) = -f(f(x))$.
Thus, $f(f(x))$ is an \boxed{odd} function.

Question:
At the 2007 Math Olympics, Team Canada won $17$ out of a possible $100$
medals. Which one of the following is closest to the fraction of medals that
 they won? $$
\frac{1}{4} \qquad \frac{1}{5} \qquad \frac{1}{6} \qquad \frac{1}{7} \qquad
\frac{1}{8}
$$

My solution:
Let's use sympy to print out the differences between the fraction of medals
that Canada won and each of the options.

```
<llm-code>
from sympy import Rational, Abs

# team Canada won 17 out of 100 medals
medal_frac = Rational(17, 100)

# list of options
options = [Rational(1, 4), Rational(1, 5), Rational(1, 6), Rational(1, 7),
Rational(1, 8)]

# let's print out the differences
[Abs(medal_frac - frac_option) for frac_option in options]
</llm-code>
<llm-code-output>
[2/25, 3/100, 1/300, 19/700, 9/200]
</llm-code-output>
Let's now check which difference is the smallest.
<llm-code>
import numpy as np

# Calculate the idx of the closest option
min_idx = np.argmin([2/25, 3/100, 1/300, 19/700, 9/200])

# Print the closest option
print(options[min_idx])
</llm-code>
<llm-code-output>
1/6
</llm-code-output>
So the answer is \boxed{1/6}.
```

Question:
A rectangular box $P$ is inscribed in a sphere of radius $r$. The surface
area of $P$ is 384, and the sum of the lengths of its 12 edges is 112. What
is $r$?

My solution:
Let the dimensions of $P$ be $x$, $y$, and $z$.
The sum of the box's edges is $4(x + y + z)$ and the surface area is $2xy +
2yz + 2xz$.
The diameter of the sphere is $2r$ and it's equal to the diagonal of the box
.
Let's now write down the equations based on the above information and solve
them using sympy.

```
<llm-code>
from sympy import symbols, Eq, solve

# define the variables
x, y, z, r = symbols('x y z r')

# equations based on the given information
eq1 = Eq(2 * (x*y + x*z + y*z), 384)
eq2 = Eq(4 * (x + y + z), 112)

# the diagonal of the box is the diameter of the sphere
diagonal_eq = Eq(x**2 + y**2 + z**2, (2*r)**2)

# solve the equations
solutions = solve((eq1, eq2, diagonal_eq), (x, y, z, r))

# let's see which values of r we get
```

```
[solution[-1] for solution in solutions]
</llm-code>
<llm-code-output>
[-10, -10, 10, 10]
</llm-code-output>
```
Since the radius of the sphere is positive, we get $r = \boxed{10}$.

Question:
A bee starts flying from point $P_0$. She flies $1$ inch due east to point $P_1$. For $j \ge 1$, once the bee reaches point $P_j$, she turns $30^{\circ}$ counterclockwise and then flies $j+1$ inches straight to point $P_{j+1}$. When the bee reaches $P_{2015},$ how far from $P_0$ is she, in inches?

My solution:
We can represent the rotation action via $\omega = e^{\pi i/6}$.
Let's assume the bee starts at the origin, so $P_{2015}$ is at the point \[z = 1 + 2 \omega + 3 \omega^2 + 4 \omega^3 + \dots + 2015 \omega^{2014}.\]
This is an arithmetic-geometric series which we can solve by simplifying the expression.
Alternatively, we can solve for |z| using sympy using the following code.
```
<llm-code>
from sympy import I, pi, exp, sqrt

# rotation action of 30 degrees
omega = exp(I * pi/6)

position = 0

for i in range(2015):
    delta = (i + 1) * omega**(i)
    position += delta

real, img = (position.as_real_imag())
# Distance from origin i.e. starting point
dist = sqrt(real**2 + img**2)
print(dist)
</llm-code>
<llm-code-output>
sqrt(2)*(1008 + 1008*sqrt(3))
</llm-code-output>
```
So the bee is $\boxed{1008\sqrt{2} + 1008\sqrt{6}}$ far from the starting point.

Question:
If $f (x) = x^2 - 1$, what is the value of $f (-1)$?

My solution:

---

