# OpenReview forum: "OpenMathInstruct-1: A 1.8 Million Math Instruction Tuning Dataset"
_NeurIPS.cc/2024/Datasets_and_Benchmarks_Track — NeurIPS 2024 Track Datasets and Benchmarks Oral_

### Official Review · Reviewer_Sz3o · 2024-07-22

**Rating:** 7
**Confidence:** 4
**Clarity:** Yes, this paper is well written.

**Review:**

This paper is good. The authors designed a reasonable math code instruction generation process with low cost by prompting an open-source model. They achieve a high coverage rate on MATH and GSM8K, and the results are promising.

The author of this paper proposes to mask important values in the question to help the model generate complete code instead of shortcuts; it is still doubtful that a small model can understand such "hints" and generate useable code. For example, the author may mask 1$\times$2=2 into A$\times$B=C, but the model may treat the A, B, and C as actual values and hardcode them into the code, or the code may have some strange logical flow, which will cause an inconsistency or the loss of instruction-following ability. But overall, it is still a good try.

The authors evaluate various models with different scales in experiments. This is enough to support the proposed dataset's ability to help the model improve its math problem-solving skills. However, it lacks more analysis on how models behave after fine-tuning the proposed dataset by case study.

**Strengths:**

- This paper is good. The authors designed a reasonable math code instruction generation process with low cost by prompting an open-source model.
- They mask text to prevent the model from generating shortcuts in the code, which is reasonable and novel.
- They achieve a high coverage rate on MATH and GSM8K, and the results are promising.

**Additional Feedback:**

N/A

**Correctness:**

Yes, the dataset construction method is good, and promising experiment results support it.

**Documentation:**

Yes, as described in Section 2.

**Limitations:**

The finetuning prompt template is static, which may lead to prompt sensitivity in real-world scenarios.

**Opportunities For Improvement:**

- Case studies on the generated code before and after text mask.
- Case studies on how models behave after fine-tuning on the proposed dataset.

**Relation To Prior Work:**

The authors have discussed related works in Section 5.

**Summary And Contributions:**

This paper proposes a code-augmented math instruction-tuning dataset, pairing each problem in MATH and GSM8K with diverse solutions. This dataset is fully generated by an open-sourced model (Mixtral 8x7b) with a well-designed process, including a few-shot hint and text mask to increase the coverage rate. Experiment results show that models from various scales fine-tuning on the proposed dataset can achieve outstanding results on almost all math-related tasks.

---

> ### Author Rebuttal · Authors · 2024-08-14
>
> We thank the reviewer for their detailed review. Below is our response to the queries raised by the reviewer:
>
> > The author of this paper proposes to mask important values in the question to help the model generate complete code instead of shortcuts; it is still doubtful that a small model can understand such "hints" and generate useable code. For example, the author may mask 1x2=2 into AxB=C, but the model may treat the A, B, and C as actual values and hardcode them into the code, or the code may have some strange logical flow, which will cause an inconsistency or the loss of instruction-following ability.
>
> Suppose the model generates solutions with these intermediate masked-out variables copied into code. In that case, there is a very good chance that this candidate solution fails to execute/or has other errors and produces an answer that is different from the ground truth answer. Since we filter our synthetic solutions, which don't lead to the correct answer of the training set questions, most likely, the solutions described by the reviewer will be filtered out. We agree with the reviewer that the model can still produce false positive solution trajectories (Figure 9 in Appendix). However, it's unclear if the use of masked solutions in the prompt exaggerates this problem.
>
> > Case studies on how models behave after fine-tuning on the proposed dataset.
>
> Unfortunately, due to space constraints, we couldn’t include the analysis of the fine-tuned model in the main text. Appendix A.2 has details of the error analysis, and Appendix C.4 presents instances from the validation set where the analyzed fine-tuned model fails.
>
> > The finetuning prompt template is static, which may lead to prompt sensitivity in real-world scenarios.
>
> Prompt robustness remains a pain point with state-of-the-art LLMs, and we agree that this will be an issue with our fine-tuned models as well due to the use of a static template. An easy fix, whose efficacy can be explored in future work, would be to generate variants of these templates (LLMs can be used for this) and dynamically use these templates in conjunction with the QA pairs.

---

### Official Review · Reviewer_vaWD · 2024-07-24
**A useful dataset for solving math problem**

**Rating:** 5
**Confidence:** 4
**Correctness:** yes.
**Clarity:** yes.

**Review:**

The method is simple and easy to follow. The experiments show that the generated dataset is of relatively high quality.

**Strengths:**

Table 3 shows the proposed dataset can help boost many models' performances.

**Additional Feedback:**

n.a

**Documentation:**

yes

**Limitations:**

It is not clear how to remove the incorrectness in the generated dataset.
The inconsistency of performance changes (increase on one dataset and decrease or no changes on others) indicates that the models learn superficial features instead of deeper math-related knowledge from the generated dataset.

**Opportunities For Improvement:**

The idea of developing a dataset which allows more flexibility than GPT-distilled ones is great. But as shown in Table 3 there is still a clear gap in performances. The ablation study which is good but can be improved to provide more insights on what actually helps and why. The analysis part is not super convincing.

**Relation To Prior Work:**

yes

**Summary And Contributions:**

This work presents the OpenMathInstruct which contains 1.8M data. The data is created by usig Mistrial to answer MATH and GSM8k.
Using the text solution in a few-shot setting, this work developed a dataset with a slightly better dataset. Experiments on the model fine tuning show that the gains in data TSC lead to better performances on the actual problem solving tasks.

---

> ### Author Rebuttal · Authors · 2024-08-17
>
> We thank the reviewer for their feedback.
>
> > The data is created by usig Mistrial to answer MATH and GSM8k.
>
> To be clear, we used the Mixtral model from Mistral. We’re not aware of any Mistrial model.
>
>
> > The ablation study which is good but can be improved to provide more insights on what actually helps and why. The analysis part is not super convincing.
>
> We would appreciate it if the reviewer could point out any missing ablations in our studies. Our ablations presented in Section 4.1 currently address: (a) a comparison of our proposed fair downsampling with naive downsampling, (b) a comparison of our proposed masked prompting vs. default prompting strategy, (c) the impact of preferring code solutions, and (d) the the effect of subject-wise prompting vs. default prompting for MATH. Due to space constraints, we present additional analysis in Appendix A, which includes the impact of data scaling and error analysis of our fine-tuned models.
>
> > It is not clear how to remove the incorrectness in the generated dataset.
>
> We developed heuristics to filter out *syntactically noisy* solutions which are presented in Section 2.3. Anecdotally, we find false positive solutions, i.e., ones using the wrong reasoning chain to arrive at the right answer, are rare in our dataset. Automatic identification of these *semantically noisy* solutions can be done via LLM as a judge or reward models. We plan to explore these techniques in future work.
>
> > The inconsistency of performance changes (increase on one dataset and decrease or no changes on others) indicates that the models learn superficial features instead of deeper math-related knowledge from the generated dataset.
>
> Given the distributional differences between the datasets, a perfect performance sync across all of them is very difficult in practice. For example, GSM8K is a grade-school dataset, while MATH is a much harder high-school competition dataset. That said, the model performances on GSM8K and MATH have a high Spearman correlation with the out-of-domain datasets presented in this paper, namely GSM-Hard, SVAMP, TabMWP, ASDiv, and MAWPS, as shown below.
>
> | OOD Datasets   | Correlation with MATH performance | Correlation with GSM8K performance |
> |----------|---------|---------|
> | GSM-Hard | 0.89    | 0.94    |
> | SVAMP    | 0.77    | 0.89    |
> | TabMWP   | 0.71    | 0.89    |
> | ASDiv    | 0.81    | 0.93    |
> | MAWPS    | 0.71    | 0.89    |
>
>
> Given these strong correlations, we believe models are not merely learning superficial math-related features.

---

### Official Review · Reviewer_XvnS · 2024-07-25

**Rating:** 9
**Confidence:** 5
**Correctness:** \
**Clarity:** The paper is well written.

**Review:**

I believe the authors of this paper have done highly significant work, and I strongly agree with the viewpoints expressed in the paper. In fact, many of the current state-of-the-art open-source models are GPT-distilled, but this approach has numerous limitations.

It is very encouraging that synthetic data is presented as a highly effective method. I look forward to seeing the authors use the ideas they propose to develop more powerful and reasoning-capable language models in the future.

**Strengths:**

- The motivation is highly valuable and the methods are very meaningful.
- The open-source model is very significant for researchers.

**Additional Feedback:**

\

**Documentation:**

\

**Ethics:**

No.

**Limitations:**

、

**Opportunities For Improvement:**

I believe the work presented in the paper is outstanding, but I have a few suggestions:

- I am very eager to see the authors update to more powerful models and look forward to your latest advancements.
- Additionally, could your methods be extended to other tasks or domains (beyond mathematical reasoning)? I think this is a worthwhile and valuable direction to explore.

**Relation To Prior Work:**

Yes.

**Summary And Contributions:**

The authors construct OpenMathInstruct-1, a high-quality math instruction tuning dataset with 1.8M problem-solution pairs. The dataset is constructed by synthesizing code-interpreter solution. They use the recently released and permissively licensed Mixtral model and then train OpenMath-CodeLlama-70B. Actually, It is a significant and inspiring work for the development of LLMs.

---

> ### Author Rebuttal · Authors · 2024-08-16
>
> We thank the reviewer for their positive feedback. Following are our responses to the reviewer’s questions:
>
> > I am very eager to see the authors update to more powerful models and look forward to your latest advancements.
>
> With the pace of model releases, we’re always playing catch up :) But we are working on it!
>
> > Additionally, could your methods be extended to other tasks or domains (beyond mathematical reasoning)? I think this is a worthwhile and valuable direction to explore.
>
> We believe many of the components developed in this paper can also be adapted to generate synthetic data for other domains. The tricky part is that, unlike mathematics, filtering data can require a more involved pipeline. LLM-based judgment and heuristics can serve as a proxy to curate the synthetic data at scale.

---

### Comment · Area_Chair_y99G · 2024-08-29
**Please engage in the author-reviewer discussion**

Dear Reviewers,

Thank you for your hard work on the papers and reviews. Please note that the deadline for the author-reviewer discussion period is approaching (August 31, 2024). Some of you have engaged in discussions with authors-thank you! For reviewers who have not yet, please discuss with authors as this is a very important part of the reviewing process and authors are eager to have further feedback from you. If there are any changes to your scores, kindly provide explanations for these adjustments.

---

### Decision · Program_Chairs · 2024-09-26

**Decision:**

Accept (Oral)

**Comment:**

The paper presents OpenMathInstruct-1, a math instruction tuning dataset with 1.8M problem-solution pairs.

Pros:
1. The dataset is well-motivated.
2. The paper designs a reasonable math code instruction generation process with low cost.
3. The created math dataset is of good quality and high coverage.